# Range, routing and kinetics of rod signaling in primate retina

**William N Grimes, Jacob Baudin, Anthony W Azevedo, Fred Rieke\***

Department of Physiology and Biophysics, University of Washington, Seattle, United States

**Abstract** Stimulus- or context-dependent routing of neural signals through parallel pathways can permit flexible processing of diverse inputs. For example, work in mouse shows that rod photoreceptor signals are routed through several retinal pathways, each specialized for different light levels. This light-level-dependent routing of rod signals has been invoked to explain several human perceptual results, but it has not been tested in primate retina. Here, we show, surprisingly, that rod signals traverse the primate retina almost exclusively through a single pathway – the dedicated rod bipolar pathway. Identical experiments in mouse and primate reveal substantial differences in how rod signals traverse the retina. These results require reevaluating human perceptual results in terms of flexible computation within this single pathway. This includes a prominent speeding of rod signals with light level – which we show is inherited directly from the rod photoreceptors themselves rather than from different pathways with distinct kinetics.
DOI: https://doi.org/10.7554/eLife.38281.001

## Introduction

Rod photoreceptors contribute to vision across a million-fold range of light intensities. At the low end of this range - for example in starlight - photons are few and far between, and retinal circuits face the considerable challenge of detecting and reliably transmitting signals resulting from the absorption of individual photons (reviewed by (*Field et al., 2005*; *Takeshita et al., 2017*)). Other challenges emerge as light levels increase. For example, in moonlight the high gain associated with detecting single photons, if unabated, would quickly saturate retinal responses. At the same time, the increased fidelity of light inputs at dawn or dusk creates opportunities for more elaborate computations than possible in starlight.

A prominent hypothesis about how retinal circuits meet these changing demands proposes that the routes that rod-derived signals take through the retina depend on mean light level (*Sharpe and Stockman, 1999*; *Tsukamoto et al., 2001*; *Bloomfield and Dacheux, 2001*; *Deans et al., 2002*) - with different circuits specialized to handle the challenges associated with different light levels. These circuits include (*Figure 1*; reviewed by (*Bloomfield and Dacheux, 2001*; *Field et al., 2005*; *Field and Chichilnisky, 2007*; *Demb and Singer, 2012*)): (1) the primary pathway, in which dedicated rod bipolar cells transmit rod signals, (2) the secondary pathway, in which gap junctions convey rod signals to cones and the associated cone circuitry, and (3) the tertiary pathway, in which Off cone bipolar cell dendrites receive direct rod input. Recent work also proposes that rod signals can reach cones (and then cone bipolar cells) through horizontal cells, creating rod-cone spectral opponency (*Joesch and Meister, 2016*).

A light-level-dependent routing of rod-derived signals has been invoked to explain several human perceptual observations (reviewed by (*Sharpe and Stockman, 1999*; *Buck, 2004*; *Stockman and Sharpe, 2006*; *Buck, 2014*; *Zele and Cao, 2014*)). For example, perceptual experiments show that the kinetics of rod-derived signals speed substantially as luminance levels increase from low to high mesopic conditions (i.e. conditions where rods and cones both contribute to vision)

**\*For correspondence:**
rieke@u.washington.edu

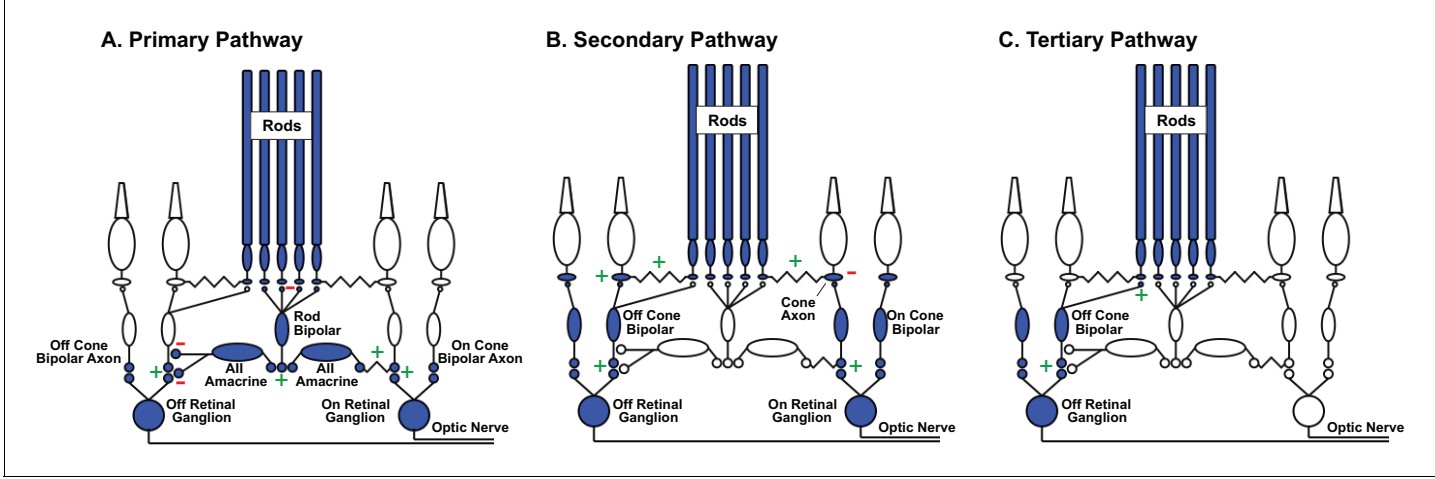

**Figure 1.** Rod signal routing in the mammalian retina. (A) In the primary pathway, rod signals are routed through dedicated rod bipolar cells to AII amacrine cells. AII amacrine cells in turn transmit 'On' signals to On cone bipolar cells through dendro-axonal gap junctions and 'Off' signals to Off cone bipolar cells through glycinergic synapses. Cone bipolar signals are subsequently transmitted to retinal ganglion cells. (B) In the secondary pathway, rod transmit signals via gap junctions to cone axons, and hence the associated cone circuitry (*Kolb, 1977*; *Schneeweis and Schnapf, 1995*; *Deans et al., 2002*; *Hornstein et al., 2005*). (C) In the tertiary pathway, rods transmit signals directly to Off cone bipolar cell dendrites (*Soucy et al., 1998*; *Hack et al., 1999*; *Tsukamoto et al., 2001*). Plus and minus signs represent sign-conserving and sign-inverting synapses.

DOI: https://doi.org/10.7554/eLife.38281.002

(*Connaughton et al., 1999*; *Sharpe et al., 1989*). Relatedly, differences in the kinetics of rod- and cone-derived signals play a central role in how these signals are combined perceptually (*MacLeod, 1972*; *Frumkes et al., 1973*; *Grimes et al., 2015*). The link between these perceptual phenomena and the routing of rod signals is based on the assumption that the primary pathway introduces larger delays in rod-derived signals than the secondary or tertiary pathways.

Work in rodent retina supports the hypothesized light-level-dependent routing of rod signals (*Soucy et al., 1998*; *Deans et al., 2002*; *Trexler et al., 2005*), although this support comes with some caveats. Specifically, while the primary pathway dominates responses of mouse ganglion cells at low light levels, the secondary pathway contributes substantially at mesopic light levels (*Deans et al., 2002*; *Ke et al., 2014*; *Grimes et al., 2014b*). The observed change in routing, however, is incomplete, with at most an approximately equal distribution of signals between primary and secondary pathways at high mesopic light levels (*Ke et al., 2014*; *Grimes et al., 2014b*). Further, there is no direct physiological evidence that the kinetics of signals traversing the primary and secondary pathways differ.

Despite its importance for understanding human vision, the routing of rod signals through the primate retina is not well understood, and it is unclear to what extent findings in rodents will translate to primate. Here, we determine (1) the range over which primate rod photoreceptors control retinal output, (2) how and when rod signals traverse a given pathway, and (3) what mechanisms shape the kinetics of rod-derived retinal outputs. Surprisingly, and unlike mouse retina, we find that rod-derived signals in primate retina are largely restricted to the primary pathway even at mesopic light levels. Under the same conditions, the responses of rods themselves speed sufficiently to explain previous human perceptual results, indicating that this speeding does not require the rerouting of rod-derived signals through faster retinal pathways.

## Results

### Rod signaling range and saturation

We start by defining the range of light levels over which rod photoreceptors respond to light inputs and comparing this range to the range over which rod-derived signals are present in the retinal outputs.

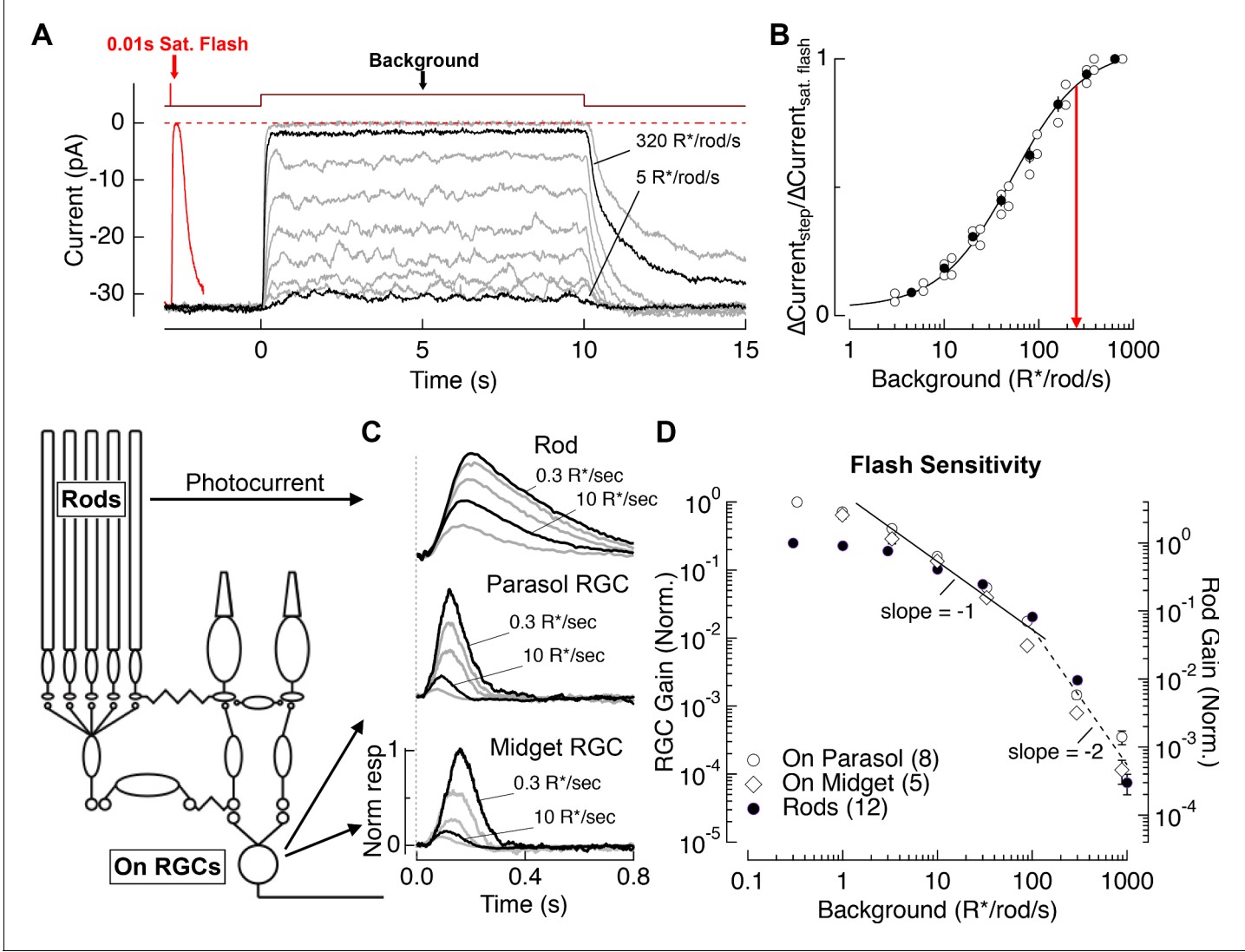

**Figure 2.** Evaluation of the range of rod signaling in phototransduction and in the retinal output. (**A**) Direct recordings of photocurrent from rod outer segments (primate). Comparison of the responses to a short-wavelength saturating flash and a family of 10 s light steps or 'backgrounds.' (B) Current suppression, relative to saturation, plotted as a function of background luminance. Red arrow (250 R*/rod/s) indicates 90% saturation. (C) Gain of rod signals probed with short wavelength flashes on a range of rod adapting backgrounds in rod photoreceptors (top), On parasol RGCs (middle) and On Midget RGCs (bottom; traces are mean responses scaled by the flash strength in R*/rod, see Materials and methods). Gain is normalized to that measured at 0.3 R*/rod/s. (**D**) Population data for rod gain measurements in rod photoreceptors and ganglion cells. The solid line shows Weber behavior, and the dashed line highlights the steep drop in gain associated with rod saturation. Gain is normalized to that measured at 0.3 R*/rod/s.

DOI: https://doi.org/10.7554/eLife.38281.003

The following source data is available for figure 2:

**Source data 1.** Excel spreadsheet with data for *Figure 2B and D*.
DOI: https://doi.org/10.7554/eLife.38281.004
**Figure Source data 2.** Matlab .mat file with data traces from *Figure 2A*.
DOI: https://doi.org/10.7554/eLife.38281.005
**Source data 3.** Matlab .mat file with data traces from *Figure 2C*.
DOI: https://doi.org/10.7554/eLife.38281.006

We used suction electrode recordings to measure rod outer segment membrane currents (i.e. phototransduction currents). *Figure 2A* compares responses to a series of light steps (10 s) with the response to a saturating flash. Light steps producing ~40 R*/rod/s halved the outer segment current, and steps producing ≥300 R*/rod/s suppressed nearly all of the dark current. *Figure 2B* collects

results from several such experiments. Mean backgrounds of 250 R*/rod/s suppressed 90% of the dark current (red line). This suppression was maintained during light steps that persisted for 2–3 min (<5% recovery of dark current, not shown). We did not test suppression for longer steps because of the lack of photopigment regeneration in our preparations (see Discussion).

To compare the signaling range of rods with that of retinal ganglion cells (RGCs; the retinal output neurons), we measured rod photocurrents and RGC spike responses to brief short-wavelength flashes across a range of backgrounds (*Figure 2C*). Spectral measurements indicated that RGC responses to these short-wavelength flashes were predominantly driven by rods for backgrounds up to 300 R*/rod/s (see Materials and methods). *Figure 2D* summarizes measurements of the background dependence of rod and RGC response gain (peak response divided by flash strength). Between darkness and 3 R*/rod/s, RGC gain declined considerably more than rod gain. This reduction in RGC gain reflects post-rod circuit adaptation (*Lee et al., 1990*; *Purpura et al., 1990*; *Dunn et al., 2006*; *Schwartz and Rieke, 2013*) – for example at the rod bipolar output synapse (*Dunn and Rieke, 2008*; *Oesch and Diamond, 2011*). For backgrounds between 3 and 100 R*/rod/ s, both rod and ganglion cell gain declined approximately inversely with background, as expected for Weber's Law (reviewed in (*Rieke and Rudd, 2009*)). Thus, for this range of light levels most of the decline in RGC gain can be accounted for by the rods. Above 100 R*/rod/s, rod signal gain in both rods and RGCs declined with increasing background more sharply than expected from Weber's Law, falling by a factor of ~100 between 100 and 1000 R*/rod/s (black line). This abrupt decline in the gain of rod-derived signals coincides closely with the suppression of outer segment current (*Figure 2B*). The reduction in rod and RGC signal gain, like the reduction in rod outer segment current, was maintained for light steps that persisted for 2–3 min.

These experiments indicate that the primary limitation on the range of rod-derived signaling in the retinal output is a decrease in the gain of rod signals, rather than decreased gain in post-rod circuitry. Rods may continue to weakly modulate retinal output signals above 300 R*/rod/s, as observed in rodents (*Yin et al., 2006*; *Naarendorp et al., 2010*; *Tikidji-Hamburyan et al., 2017*), but such responses are small compared to cone-derived responses for all but short wavelength stimuli under our experimental conditions (see Materials and methods). Hence, we refer to the sharp drop in sensitivity above 300 R*/rod/s as rod saturation and focus on rod signaling for backgrounds of 0 to 300 R*/rod/s. Note that 300 R*/rod/s is an upper bound for large rod-derived RGC responses under the conditions of our experiments, and that cone-derived responses could dominate at lower light levels in pathways with relatively weak rod-derived signals.

## Rod signal routing

What route do rod signals take through the primate retina across the range of light levels identified above? The results described below show, surprisingly, that signaling through the rod bipolar pathway dominates rod-derived RGC responses across a broad range of stimuli and light levels.

We start by comparing responses of key cells in the primary and secondary rod pathways. One such cell is the AII amacrine cell. AII amacrines receive direct glutamatergic input from rod bipolar cells through AMPA-type glutamate receptors and make bidirectional electrical synapses with On cone bipolar cell axons (*Figure 3A*). Thus, under control conditions, AII responses reflect signaling from both the primary pathway via rod bipolar cells and the secondary pathway via gap junctional input from cone bipolar cells. In the presence of AMPA receptor antagonists, remaining AII responses reflect input only from the secondary pathway (*Cohen, 1998*; *Trexler et al., 2001*; *Murphy and Rieke, 2008*; *Münch et al., 2009*; *Grimes et al., 2014b*; *Ke et al., 2014*). Because AMPA receptor antagonists likely alter retinal signaling in multiple ways, we performed complementary experiments on horizontal cells that did not require pharmacological manipulation. We start by describing the AII results.

After achieving a stable current-clamp recording, short-wavelength flashes were delivered on backgrounds ranging from darkness to 2000 R*/rod/s (*Figure 3B*). These backgrounds extend beyond rod saturation (see *Figure 2*). Flash strength scaled with background, so that the contrast was fixed (e.g. flashes produced 0.07 R*/rod from a background of 1 R*/rod/s and 0.7 R*/rod from a background of 10 R*/rod/s). Primate AII amacrines did not exhibit the large steady-state hyperpolarization with backgrounds seen in mouse AII amacrines (*Grimes et al., 2014b*). The sensitivity of AII flash responses recorded in control conditions closely matched that of excitatory synaptic inputs to On parasol RGCs (*Figure 3—figure supplement 1*). This is consistent with the known role of AII

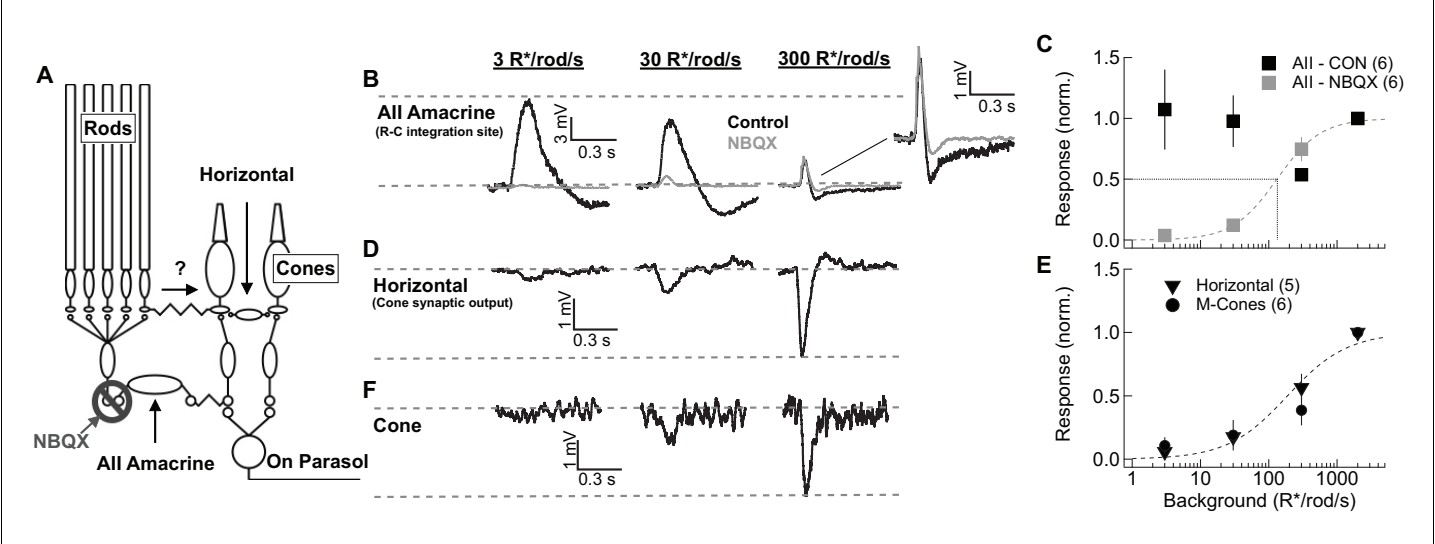

**Figure 3.** Comparison of rod signal strength in cells of the primary and secondary rod pathways (primate). (**A**) Schematic of the primary and secondary rod pathways and action of the pharmacological manipulation used in the AII amacrine recordings. (**B,D,F**) Responses to short wavelength 600% contrast flashes across a range of rod backgrounds in AII amacrine cells, H1 horizontal cells, and M-cones. (**B**) Voltage responses of a current-clamped AII amacrine cell before (black) and during exposure to NBQX (10 µM). (**C**) Population data for AII amacrine recordings with and without NBQX. Responses are normalized by the response amplitude measured at 2000R*/rod/s. (**D**) Voltage responses of a current-clamped H1 horizontal cell. (**E**) Population data for H1 horizontal and M-cone recordings. Responses are normalized by the response amplitude measured at 2000 R*/rod/s. (**F**) Voltage responses of a M-cone recorded in the perforated-patch configuration.

DOI: https://doi.org/10.7554/eLife.38281.007

The following source data and figure supplements are available for figure 3:

**Source data 1.** Excel spreadsheet with data for *Figure 3C and E*.
DOI: https://doi.org/10.7554/eLife.38281.011
**Figure supplement 1.** Rod signals in AII amacrine cells are blocked by NBQX (10 µM) at low light levels.
DOI: https://doi.org/10.7554/eLife.38281.008
**Figure supplement 2.** Rod responses measured in cones.
DOI: https://doi.org/10.7554/eLife.38281.009
**Figure supplement 3.** Rod signals are weak in H1 horizontal cells.
DOI: https://doi.org/10.7554/eLife.38281.010

amacrine cells in the circuits that convey rod signals to RGCs across light levels and confirms that AII responses are a reliable indicator of the strength of rod signals in On retinal circuits.

Block of AMPA receptors with NBQX reduced AII responses more than 80% at backgrounds at or below 30 R*/rod/s (*Figure 3B–C*). Several properties of the NBQX-insensitive responses suggest that they originated in cones. First, they had shorter durations than control AII responses (*Figure 3—figure supplement 1*). Second, they grew with increasing background, reaching half-maximal amplitude at ~300 R*/rod/s (*Figure 3C*; note that because of the scaling of flash strength with background, a constant response does not mean responses are saturated but instead that they are proportional to contrast). Most importantly for our purposes, NBQX-insensitive responses were largely absent over the range of light levels in which rods modulate the retinal outputs (i.e. below 300 R*/rod/s). This correspondence between rod saturation (*Figure 2*) and the emergence of NBQX-insensitive responses (*Figure 3*) suggests that the secondary rod pathway does not play a major role in transmitting rod signals to ganglion cells in primates, regardless of light level.

The dominance of the primary rod pathway suggested by the AII recordings is subject to the caveat that NBQX will impact retinal signaling in multiple ways. To test this conclusion without the need for pharmacology, we made current-clamp recordings from H1 horizontal cells. H1 horizontal cells receive direct synaptic input from L- and M- cones (*Figure 3A*; [*Kolb, 1970*; *Rodieck, 1998*; *Verweij et al., 1999*]). Rod input could reach H1 horizontal cells through rod-cone gap junctions and cone synaptic output (*Nelson, 1977*; *Schneeweis and Schnapf, 1995*; *Hornstein et al., 2005*) or through direct rod contacts onto H1 cells as suggested by recent work in mouse (*Joesch and*

*Meister, 2016*). H1 horizontal cells provide a readout of signals in either of these alternate pathways. H1 responses, like the NBQX-insensitive responses of AII amacrine cells, were weak at low backgrounds but became pronounced at backgrounds ≥ 300 R*/rod/s (*Figure 3D,E*). Responses of cones were very similar to those of horizontal cells, with the emergence of a sizeable response only for backgrounds ≥ 300 R*/rod/s (*Figure 3E,F*). Rod responses measured in cones and horizontal cells in our experiments were similar in magnitude and kinetics to those in previous studies (*Hornstein et al., 2005*; *Verweij et al., 1999*) (*Figure 3—figure supplement 2*).

In a subset of horizontal cell recordings, we compared responses to short-wavelength (rod-preferring) and long-wavelength (cone-preferring) flashes (*Figure 3—figure supplement 3*). At backgrounds below 300 R*/rod/s, responses to short wavelength flashes had substantially slower kinetics than responses to long wavelength flashes, consistent with rod signaling. These responses, however, were small even for flashes that elicited large RGC responses. At backgrounds ≥ 300 R*/rod/s, responses to long- and short-wavelength flashes had very similar kinetics, suggesting that at these light levels both responses originated in the cones.

The weak rod responses in cones and horizontal cells are consistent with the NBQX-insensitive responses in AII amacrine cells. The consistency of these results mitigates concerns about off-target effects of AMPA receptor block. Thus, these results collectively indicate that the secondary pathway plays, at most, a minor role in transmitting rod signals to ganglion cells in primate retina for the stimuli probed here.

## Comparison of rod signal routing in mouse and primate

The lack of prominent secondary pathway contributions in primate was unexpected, as previous experiments in rodents have assigned a substantial role to the secondary pathway in transmitting rod signals (*Deans et al., 2002*; *Trexler et al., 2005*; *Grimes et al., 2014b*; *Ke et al., 2014*). To confirm this interspecies difference, we repeated several experiments from *Figures 2* and *3* in mouse retina (*Figure 4*).

Suction recordings from rod outer segments were again used to estimate the range of rod signaling by comparing the suppression of current produced by a light step with that produced by a saturating flash, as in *Figure 2A*. Near-complete suppression of the outer segment current occurred at higher light levels in mouse compared to primate rods (90% saturation at 1000 vs 250 R*/rod/s; *Figure 4A*). Voltage responses of AII amacrine cells in NBQX (*Figure 4B*) and of horizontal cells in control conditions (*Figure 4C–D*) to fixed-contrast stimuli indicated that the contrast sensitivity of signals in the secondary pathway in mouse was half maximal at ~5 R*/rod/s, nearly two orders of magnitude lower than the half-maximal intensity of secondary-pathway rod signals in primate. For backgrounds ≥ 50 R*/rod/s, responses of mouse horizontal cells and NBQX-insensitive responses of mouse AII amacrines to fixed-contrast stimuli depended weakly on background. This does not mean that responses are saturated; instead, consistent responses to fixed contrast across backgrounds indicate Weber behavior (i.e. that gain scales inversely with background).

These experiments indicate that signaling in mouse and primate rod photoreceptors is much more similar than signaling in the circuits that convey rod signals to ganglion cells. Specifically, the secondary pathway in mouse begins contributing to retinal output at light levels that are more than 100-fold below rod saturation, whereas in primate signals from the secondary pathway are largely absent below rod saturation.

## Routing of responses to continuous stimuli

We next considered whether continuous stimuli might elicit larger signals through the secondary pathway than the brief flashes used thus far. Sinusoidal stimuli are of particular relevance since they are used in many of the human perceptual studies that motivate the light-level-dependent rod routing hypothesis.

We started by recording excitatory synaptic input to an On parasol RGC in response to short- (rod-preferring) and long- (cone-preferring) wavelength sinusoidal stimuli at a mid-mesopic light level (10 R*/rod/s). The contrasts of the short and long wavelength stimuli were adjusted so that when modulated individually at 2 Hz they produced roughly equal amplitude modulation of the RGC's excitatory inputs. These contrast amplitudes were then held fixed as we explored responses to a range of temporal frequencies (*Figure 5A,B*). As observed previously (*Dacey, 2000*), cone-

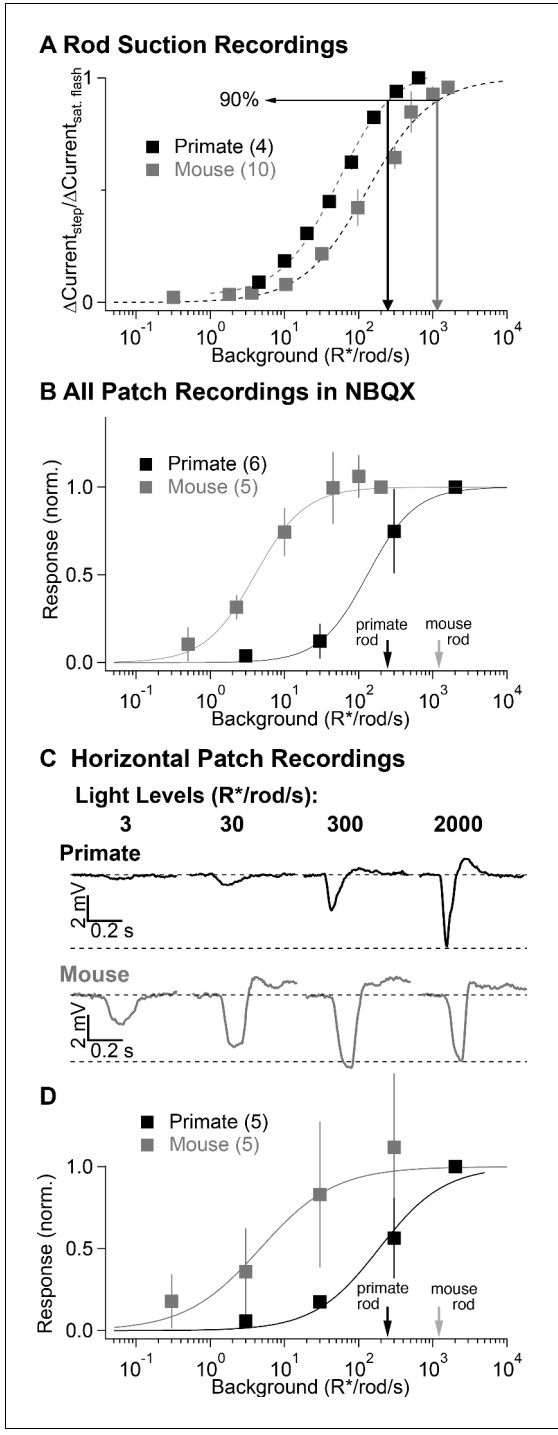

**Figure 4.** Comparison of secondary pathway activity in mouse and primate under identical experimental conditions. (**A**) Steady-state measurements of the dependence of rod current on mean light level from suction recordings in primate and mouse as in *Figure 2A*. Traces are normalized to the peak amplitude of a brief saturating flash. (**B**) NBQX-insensitive responses to fixed contrast flashes across backgrounds in AII amacrine cells recorded in current-clamp (as in *Figure 3*). Asymptotic responses in these experiments (and in C-D) represent Weber behavior, not signal saturation. Black and gray arrows indicate primate and mouse 90% rod saturation levels. (**C**) Horizontal cell current-clamp recordings to short wavelength flashes across a range of rod adapting backgrounds in primate (top) and mouse (bottom) as in *Figure 3*. (**D**) Population data from horizontal cell recordings in primate and mouse, normalized to the responses recorded at 2000 R*/rod/s.
DOI: https://doi.org/10.7554/eLife.38281.012

*Figure 4 continued on next page*

*Figure 4 continued*

The following source data is available for figure 4:

**Source data 1.** Excel spreadsheet with data for *Figure 4A, B and D*.

DOI: https://doi.org/10.7554/eLife.38281.013

derived responses were much larger than rod-derived responses at temporal frequencies $\geq$ 10 Hz (*Figure 5B*).

We next recorded from H1 horizontal cells in the same piece of retina and measured responses to the same stimuli (i.e. contrasts and background) used in the parasol recordings. Across all temporal frequencies that elicited clear rod responses ($\leq$10 Hz), the ratio of H1 responses to rod- and cone-preferring stimuli was at least seven times smaller than that for On parasol responses to the same stimuli (*Figure 5C,D*). Hence, rod-derived signals recorded in the secondary pathway in response to sinusoidal stimuli are too weak to explain rod-derived RGC responses. Instead, the primary rod pathway dominates responses to both sinusoidal and flashed stimuli.

A particular perceptual observation that motivates the rod routing hypothesis is an unexpectedly low sensitivity to 15 Hz rod flicker over a specific luminance range (*Sharpe et al., 1989*). A proposed explanation for this percept is that rod signals in slow and fast retinal circuits arrive at a downstream integration site out of phase and hence cancel (*Sharpe and Stockman, 1999*). The results described above are inconsistent with this proposal, but also do not probe the relevant stimuli directly. Hence, we measured parasol spike responses to 15 Hz flicker for light levels from 1 to 300 R*/rod/s (and 2000 R*/rod/s to measure cone-derived responses). This includes the light levels at which the perceptual rod flicker null occurs (~5 R*/rod/s; [*Sharpe et al., 1989*]). On parasol responses increased monotonically without an apparent null or abrupt change in phase needed to explain the perceptual observations (*Figure 5—figure supplement 1A*). More complete frequency tuning curves measured across the same range of light levels showed a gradual increase in responses at high frequencies with increasing light level from 1 to 300 R*/rod/s (*Figure 5—figure supplement 1B*). In particular, temporal tuning did not exhibit any abrupt discontinuities across this range of light levels suggestive of rod-rod cancelation. This data suggests that the perceptual flicker null does not originate from convergence of parallel rod circuits prior to On parasol RGCs (see Discussion and *Cao et al., 2010*).

## Routing of rod signals in Off retinal circuits

Our results thus far suggest that the primary rod pathway continues to convey rod signals to On parasol ganglion cells even when the rods are approaching saturation. Does the primary pathway also dominate responses in Off retinal circuits? Rod signals could reach Off cone bipolar cells from three known sources (*Figure 1*): (1) dendritic input directly from rods (i.e. tertiary pathway), (2) dendritic input from cones (i.e. secondary pathway), and (3) axonal inhibitory input from AII amacrine cells (i.e. primary pathway). The wiring differences between On and Off circuits suggests that both the relative weighting of rod- and cone-derived signals and the routing of rod signals could differ.

To test these hypotheses, we first compared the relative weighting of rod- and cone-derived signals in the excitatory synaptic inputs to On and Off parasol and midget RGCs at a mean light level of 20 R*/rod/s (*Figure 6A–C*). Like the experiments in *Figure 5*, we began by adjusting the contrasts of rod- and cone-preferring stimuli so that they produced equal amplitude responses in an On parasol RGC (*Figure 6A,C*). After achieving a match, we presented these response-equated stimuli while recording from other RGC types in the same piece of retina. In Off RGCs, responses to rod-preferring stimuli were roughly half as large as responses to cone-preferring stimuli (*Figure 6B,C*). This difference in the relative sensitivity of rod and cone signals indicates in turn that rod signals are less strongly routed through Off cone bipolar cells than On cone bipolar cells at this light level. This difference in weighting will be an important consideration for how different retinal circuits contribute to perception at these light levels.

The On/Off asymmetry supports the conclusion that the secondary rod pathway does not convey strong rod-derived signals in primate. In the secondary pathway, rod- and cone-derived signals are mixed prior to transmission to On and Off bipolar cells; such mixing should lead to similar weighting in On and Off circuits.

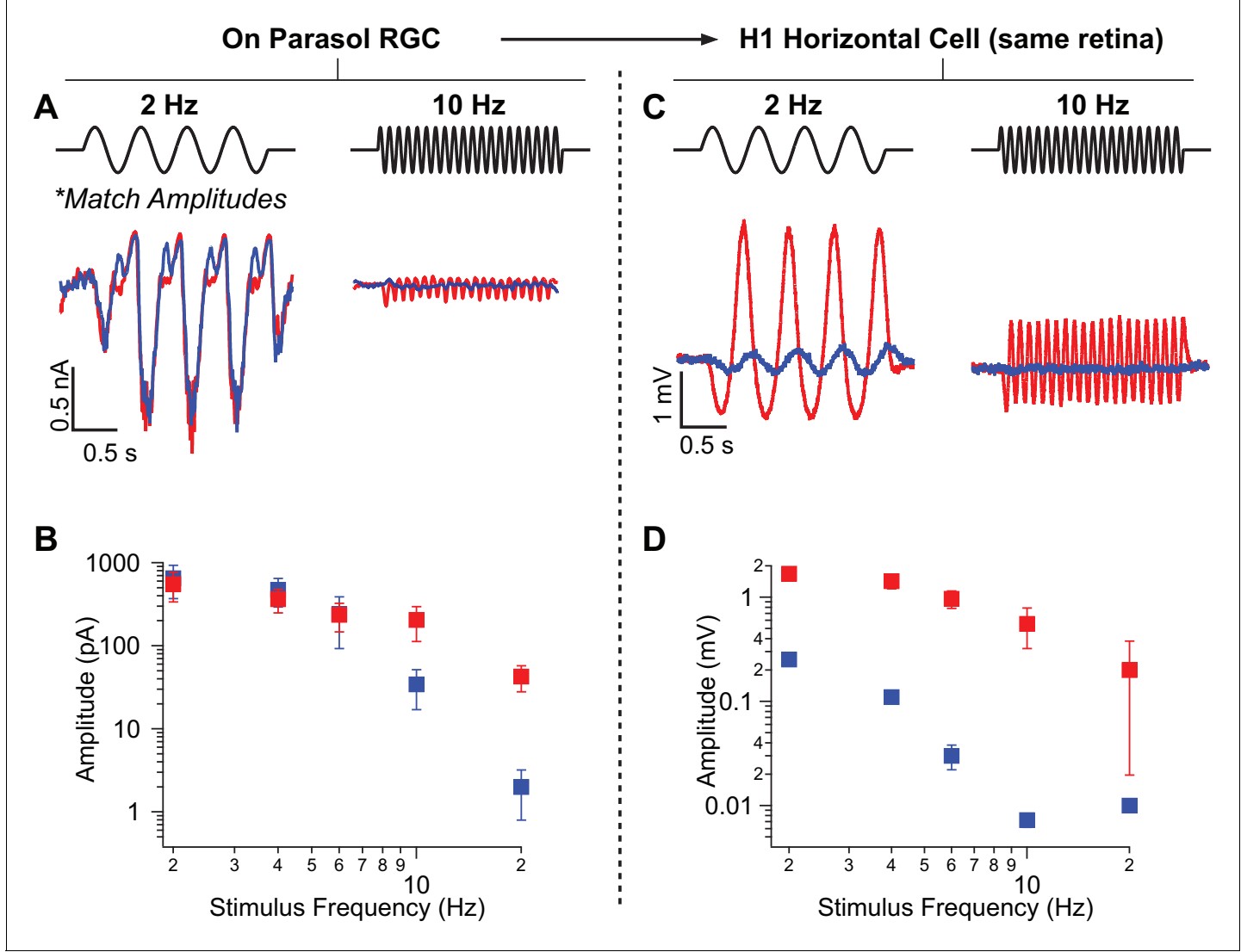

**Figure 5.** Rod-signals generated by sine-wave stimuli are also restricted to the RB pathway at backgrounds of ~20 R*/rod/s and ~200 R*/L cone/s. (**A**) Excitatory synaptic input recorded from an On parasol RGC in response to sinusoidally-modulated short (rod-preferring; blue traces) and long (cone-preferring; red traces) wavelength stimuli. Contrasts were adjusted to produce equal modulation at 2 Hz and were then held fixed for all subsequent recordings (e.g. different frequencies, horizontal recordings). (**B**) Population data from On parasol RGC recordings (n=2) plotting response modulation versus stimulus frequency. (**C**) Voltage response of an H1 horizontal cell (from the same retinal mount) to short and long wavelength stimuli for the same contrast used for the parasol cell in A. (**D**) Population data from H1 horizontal recordings (n=4) plotting response modulation versus stimulus frequency.

DOI: https://doi.org/10.7554/eLife.38281.014

The following source data and figure supplement are available for figure 5:

**Source data 1.** Excel spreadsheet with data for *Figure 5B and D*.
DOI: https://doi.org/10.7554/eLife.38281.016
**Figure supplement 1.** Responses to 100% contrast short wavelength LED modulation across stimulus frequency and background.
DOI: https://doi.org/10.7554/eLife.38281.015

Rod-derived responses in Off RGCs could also arise through the tertiary pathway via direct rod input to Off cone bipolar cells (*Figure 6D*). To test this possibility, we used a mixture of a mGluR6 agonist (APB) and antagonist (LY341495) to suppress synaptic input to all On (both rod and cone) bipolar cells. This pharmacological manipulation substantially (but not completely) suppresses light-dependent changes in excitatory synaptic input to On parasol cells (*Ala-Laurila et al., 2011*), and hence should suppress input to Off RGCs originating from the primary pathway. However, LY/APB

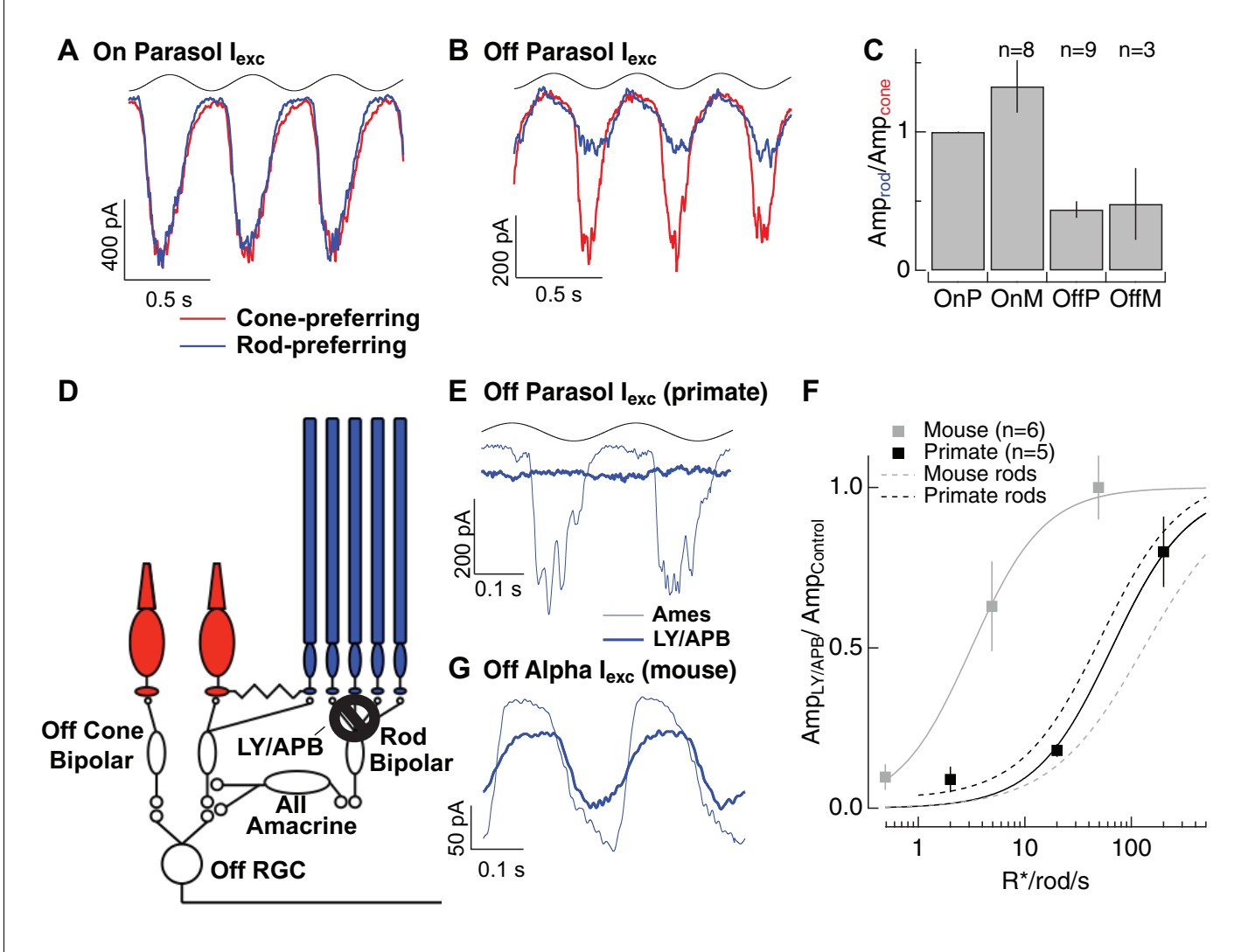

**Figure 6.** Rod signals in the tertiary rod pathway are weak in primate retina. (**A**) Excitatory synaptic input recorded from an On parasol RGC in response to sinusoidally modulated short (rod-preferring; blue traces) and long (cone preferring; red traces) wavelength stimuli. Contrasts were adjusted to produce equal modulation at 2 Hz, and were then held fixed for subsequent recordings from other cell types (e.g. Off parasol RGC). (**B**) Excitatory synaptic input recorded from an Off parasol RGC from the same retinal mount as A and in response to the same stimuli. (**C**) Relative weighting of rod and cone signals in excitatory inputs to On and Off RGCs. Mean light levels for A-C were~20R*/rod/s and~200R*/L-cone/s. (**D**) Schematic of the primary and tertiary rod circuits that influence Off cone bipolar signaling and the actions of the mGluR6 agonist/antagonist mixture LY/APB. (**E**) Rod-derived excitatory synaptic inputs to an Off parasol cell at 20 R*/rod/s in control conditions and after suppressing activity in all On bipolar cells with an mGluR6 agonist/antagonist cocktail (LY/APB, see Materials and methods). (**F**) Response ratio (cocktail:control) across cells as a function of mean luminance. Data are plotted as mean ± SEM. (**G**) Rod-derived excitatory inputs to an Off alpha RGC (mouse; 5 R*/rod/s) in control conditions and after suppressing activity in all On bipolar cells with an mGluR6 agonist/antagonist cocktail (LY/APB, see Materials and methods).
DOI: https://doi.org/10.7554/eLife.38281.017

The following source data and figure supplement are available for figure 6:

**Source data 1.** Excel spreadsheet with data for *Figure 6C and F*.
DOI: https://doi.org/10.7554/eLife.38281.019

**Figure supplement 1.** Rod, but not cone, signals in excitatory inputs to Off parasol RGCs arise from the primary rod pathway under mesopic conditions.
DOI: https://doi.org/10.7554/eLife.38281.018

should leave intact both secondary and tertiary pathway input to Off cone bipolar cells. Incomplete suppression of signaling in On circuits in these experiments would cause us to overestimate the contributions of non-primary pathway signaling.

Suppression of On pathways had little effect on the excitatory responses to long wavelength stimuli (*Figure 6—figure supplement 1*), indicating, as expected, that direct cone input to Off cone bipolar dendrites remained intact. However, suppression of On pathways reduced responses to short wavelength stimuli in Off parasol cells more than 80% at backgrounds $\leq$ 20 R*/rod/s (*Figure 6E,F*). LY/APB reduced Off parasol excitatory synaptic input considerably less at higher light levels (20% reduction at 200 R*/rod/s). Note, however, that we expect cone-derived signals to emerge at lower light levels in Off cells compared to On cells due to the difference in weighting summarized in *Figure 6A–C*, and hence cones are likely to contribute substantially to responses of Off RGCs at 200 R*/rod/s. Although these results are subject to the caveat that the pharmacology could have off-target affects, they suggest that rod signals reach Off RGCs mainly through the primary rod pathway over most or all of the mesopic range.

We repeated the LY/APB experiments in mouse retina (*Grimes et al., 2014a*) to provide a direct comparison across species (*Figure 6G,H*; see Materials and methods for controls). At a background of 0.5 R*/rod/s, LY/APB reduced excitatory inputs to Off sustained alpha RGCs by ~90%, indicating that the primary pathway dominates signaling at this light level. At a background of 5 R*/rod/s, LY/APB reduced responses by 35%, and at a background of 50 R*/rod/s, responses were little affected. This is consistent with the results from *Figure 4* that show that sizable rod signals reach AII amacrine cells through routes other than the primary pathway in mouse retina. The LY/APB insensitive responses in mouse Off RGCs could similarly arise via the secondary pathway and/or from the tertiary pathway; both of these alternative pathways can convey substantial rod signals at mesopic light levels in this species.

Collectively, the experiments in *Figures 3–6* indicate a surprising difference between how rod signals traverse the mouse and primate retinas. Specifically, unlike the situation in mouse, the primary rod pathway provides the dominant route that rod-derived signals take through the primate retina across scotopic and mesopic light levels for a wide range of stimuli. This dominance of the primary pathway means that perceptual phenomena previously attributed to a change in routing need to be reinterpreted (e.g. section on kinetics below); it also maximizes opportunities to independently process rod and cone signals since they are not mixed until late in the retinal circuitry.

## Rod signal kinetics

Perceptual experiments show that the kinetics of rod-derived signals speed relative to cone-derived signals as light levels increase (*Sharpe et al., 1989*). This speeding is often attributed to a luminance-dependent change in the dominant route that rod-derived signals take through the retina (reviewed by (*Buck, 2004*; *Chen et al., 2000*; *Stockman and Sharpe, 2006*; *Buck, 2014*)), but the experiments described above suggest that such rerouting does not occur. Instead, as described below, the shift in kinetics of rod-derived signals appears to originate within the rods themselves.

We first determined whether responses of RGCs under our experimental conditions exhibited kinetic shifts similar to those observed perceptually. Spikes (*Figure 7A*) and excitatory synaptic inputs (*Figure 7B*) were recorded from On parasol RGCs in response to sinusoidal modulation across a range of light levels (0.2–100 R*/rod/s) over which rods dominate RGC responses. Both spike responses and excitatory synaptic inputs sped by ~100 ms across the range of light levels probed (*Figure 7C*). This speeding was similar for temporal frequencies of 4–10 Hz (data not shown). Kinetic changes between 1 and 10 R*/rod/s were similar in magnitude to those measured in perceptual experiments (*Sharpe et al., 1989*), suggesting that the retina makes a substantial contribution to the speeding observed perceptually. The conserved routing of rod signals across these light levels indicates that these kinetic shifts occur within elements of the primary rod pathway rather than from a change in the dominant pathway conveying rod signals through the retina.

We returned to rod suction recordings to determine if rod phototransduction contributes to the kinetic shifts in the retinal output. Measuring rod responses to sinusoidal stimuli for light levels below 1 R*/rod/s would be near impossible due to the noise associated with stochastic photon absorption. Instead, we measured responses to brief flashes (*Figure 7D*) and used these to infer responses to sinusoids (see Materials and methods). Flash and inferred sinusoidal responses sped as light levels increased (*Figure 7F*), but not to the full extent observed in responses of RGCs. Specifically, the

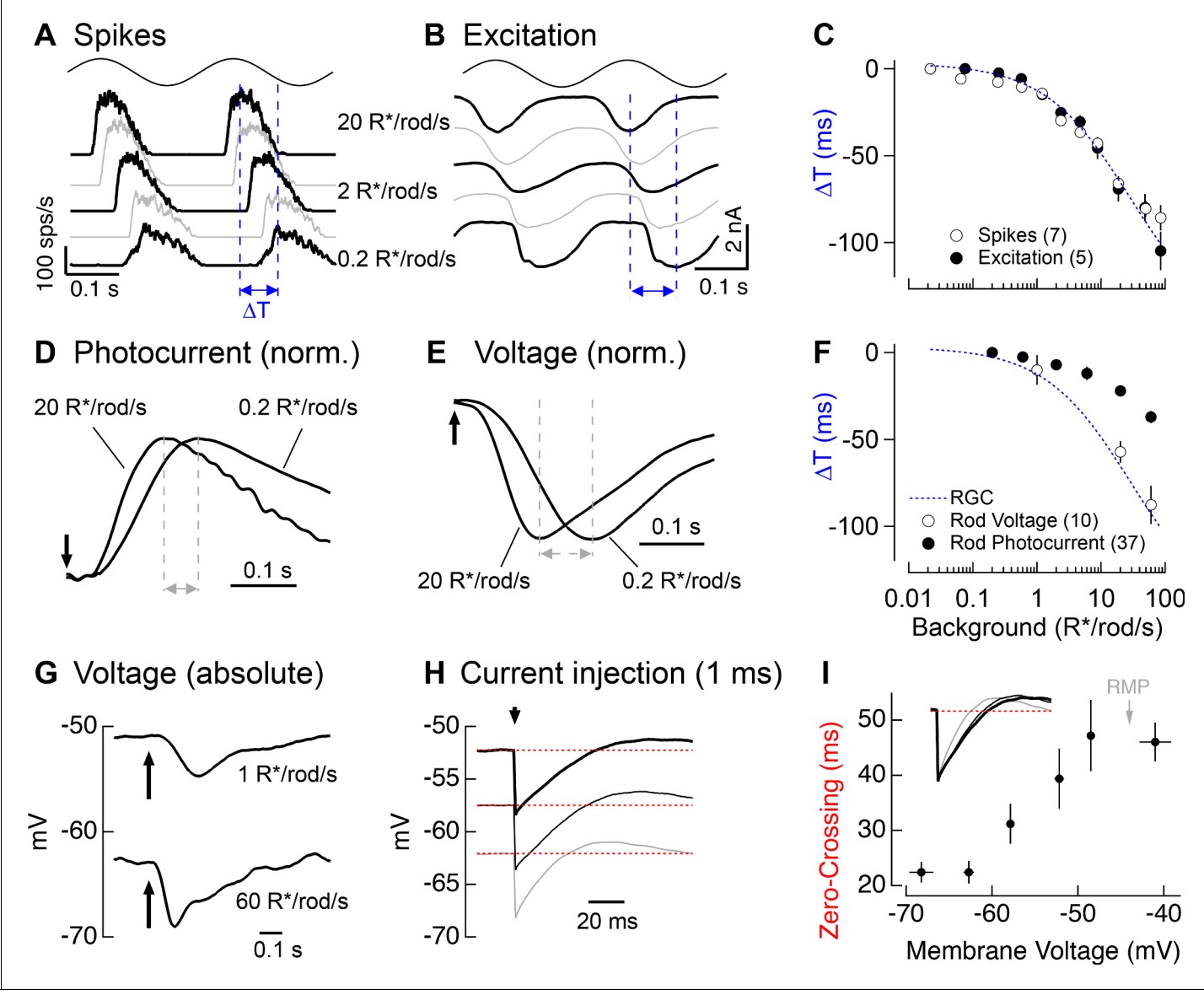

**Figure 7.** Change in kinetics of rod-derived responses across light levels. (A) Spike responses to 4 Hz sinusoidal stimuli recorded from an On parasol across a range of rod backgrounds. (B) Excitatory input currents to the same 4 Hz stimuli. (C) Temporal shifts in On parasol responses plotted versus background luminance. The smooth empirical fit to the data is a Hill curve, which is reproduced in F. (D) Recordings of rod outer segment photocurrents in response to brief flashes for two rod backgrounds (normalized). (E) Whole-cell current-clamp recordings from rod photoreceptors to brief flashes for the same two backgrounds as D (normalized). (F) Temporal shifts in rod photoreceptor recordings plotted versus rod background. (G) Exemplary current-clamp recording of a rod photoreceptor responding to a brief flash on two different rod backgrounds. Rods exhibited large steady-state hyperpolarizations as luminance levels increased. (H) Membrane response of a rod to a 1 ms current injection recorded at various physiological membrane potentials. Red dotted lines show bas(I) Measured zero-crossing time of rod responses (n=9) to 1 ms current injections plotted as a function of baseline membrane potential.

DOI: https://doi.org/10.7554/eLife.38281.020

The following source data is available for figure 7:

**Source data 1.** Excel spreadsheet with data for *Figure 7C,F and I*.

DOI: https://doi.org/10.7554/eLife.38281.021

change in kinetics observed in rod phototransduction currents could account for ~40% of the change in kinetics observed in RGC responses (*Figure 7F*, compare dashed blue line and closed circles).

Recent work on current-to-voltage transformations in mouse and goldfish photoreceptors indicates that intrinsic membrane conductances can speed visual responses in rods and cones (*Sothilingam et al., 2016*; *Seeliger et al., 2011*; *Della Santina et al., 2012*; *Howlett et al., 2017*). Could a similar mechanism account for the additional speeding between rod outer segment currents and RGC responses? To answer this question, we made whole-cell current-clamp recordings from rod photoreceptors in retinal slices. To avoid rundown of responses, we focused on a limited number of backgrounds (typically 2–3 per recording). Rod voltage responses (*Figure 7E*) to brief flashes showed larger changes in kinetics than observed in the photocurrents (*Figure 7D*). Further, when we used the flash responses to predict responses to sinusoidal stimuli, we found that they could fully explain the ~100 ms change in kinetics of RGC responses to sinusoidal stimuli (*Figure 7F*, dashed blue line and open circles).

The speeding of the rod voltage response coincided with a large steady-state hyperpolarization of the rod membrane potential ($9.0 \pm 0.7$ mV upon a step from 1 to 60 R*/rod/s, n = 10; *Figure 7G*). To test whether this hyperpolarization could alter the rod membrane time constant, we measured responses to brief hyperpolarizing current pulses across the physiological range of membrane voltages (with the voltage set by injecting steady currents; *Figure 7H*). Hyperpolarization indeed sped the membrane time constant by a factor of ~2 (*Figure 7I*), and this effect was well-matched to the physiological range of rod voltages (resting membrane potential in darkness of $-44 \pm 1$ mV, n = 10).

These experiments indicate that the change in kinetics of the rod-derived retinal outputs originates largely from the rods themselves rather than from a light-dependent shift in the routing of rod signals through fast and slow retinal circuits. Background-dependent changes in post-rod retinal signaling could further shape the kinetics of responses; such effects, however, appear relatively minor as the speeding of rod-derived RGC responses can be explained by changes in the kinetics of phototransduction together with voltage-dependent changes in the conversion of outer segment currents to voltages.

## Discussion

### Meeting the challenges of mesopic vision

Sensory systems face the considerable challenge of encoding and processing widely varying inputs. Retinal processing across the 24 hr light-dark cycle exemplifies these challenges. Near visual threshold (e.g. starlight), photons arrive at individual rods only once every hundred to thousand seconds on average. Reliably detecting these sparse inputs requires integration of signals across many rods and across time. Under mesopic conditions (e.g. dawn or dusk), the rate at which photons arrive at the retina increases as much 100,000-fold; this increased rate of photon arrivals alters the functional challenges and opportunities facing retinal circuits. For example, controlling the gain of rod-derived signals to avoid saturating retinal responses becomes a dominant consideration, and the improved signal-to-noise ratio of the inputs enables computations that are not possible in starlight.

A long-standing hypothesis about how rod vision operates effectively from starlight to twilight is that rod signals are routed through different retinal circuits under different conditions, such that the amplification and filtering properties of the dominant circuit are matched to the properties of the input (*Dunn and Rieke, 2006*). Thus, mechanisms in the primary or rod bipolar pathway are well suited to amplify single-photon responses and remove noise (*Field and Rieke, 2002b*; *Grimes et al., 2014a*) and hence support vision in starlight. As light levels increase, the gain of rod signals measured in the retinal output decreases considerably (*Lee et al., 1990*; *Purpura et al., 1990*; *Schwartz and Rieke, 2013*), reflecting adaptation in both the rod photoreceptors and post-rod retinal circuits (*Dunn et al., 2006*; *Dunn and Rieke, 2008*; *Oesch and Diamond, 2011*).

In addition to decreased gain, RGC responses speed considerably with increasing light level (*Figure 8A* insets). Indeed, the integration time of rod-derived RGC responses decreases by a factor of ~3 from low scotopic to high mesopic light levels. The result is a matching of temporal integration (*Figure 8A*) and signal-to-noise ratio (*Figure 8B*) to the statistical fluctuations in the light input; the open symbols in *Figure 8B* show how signal-to-noise depends on mean light level for a fixed (short) integration time, whereas the closed symbols show the situation with a changing integration time. Retinal circuits integrate for relatively long times at low light levels, where detecting sparse photon absorptions is a primary challenge. At low light levels, signal-to-noise is improved ~2-fold compared

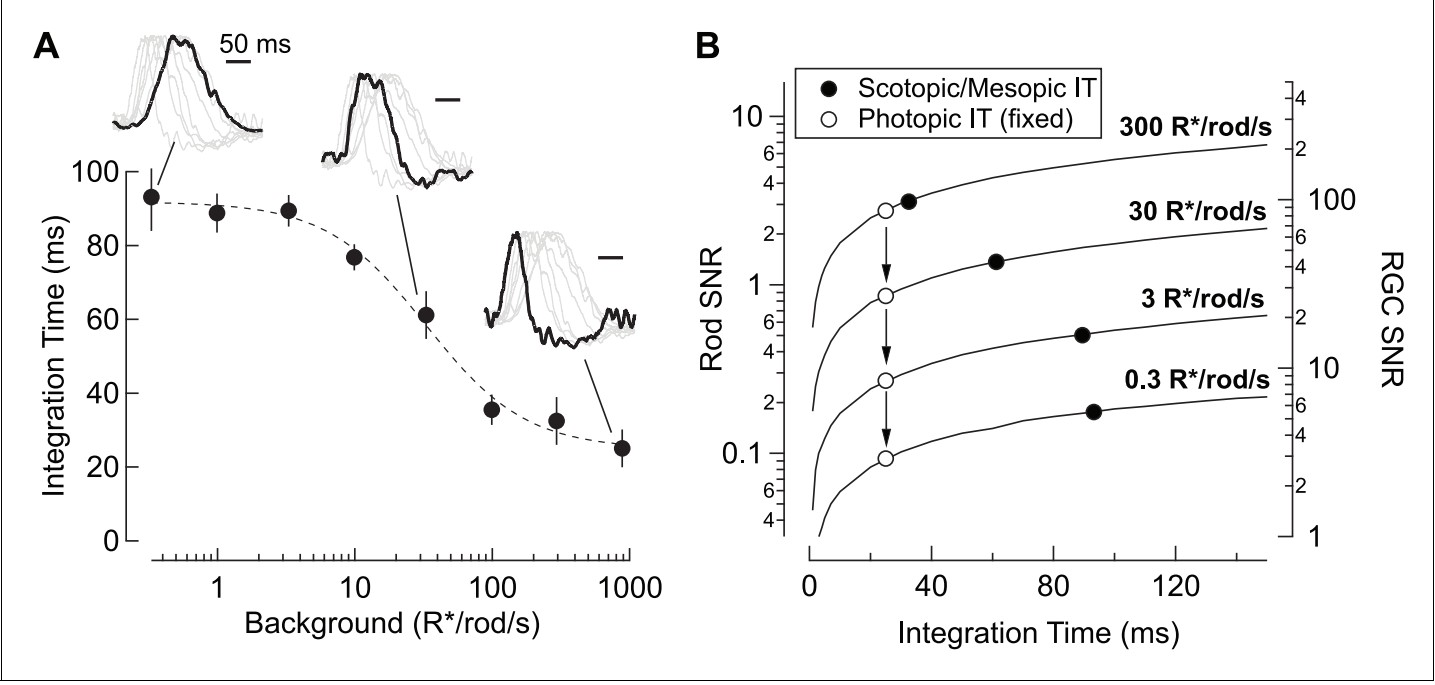

**Figure 8.** Changes in rod signal kinetics: tradeoffs between timing and SNR. (**A**) Integration time of retinal output measured from On parasol RGCs (n=8) as a function of background luminance. Integration time corresponds to the integral of the response (up to the zero-crossing) divided by the peak amplitude. Inset traces show normalized PSTHs of spike responses from On parasol RGCs, with three luminance levels (i.e. 0.3, 30 and 900 R*/rod/s) highlighted. (**B**) Model simulating the SNR for photon capture for a range of integration times, and for various levels of mean luminance (i.e. photon flux). Longer integration times improve the reliability of retinal encoding. Solid lines represent the SNRs experienced by individual rods (left axis) and individual RGCs (right axis) as a function of integration time (taken from A) for a range of photon capture rates (i.e. luminance). Black markers represent integration time measurements taken from On parasol RGC spike recordings. White markers represent a simulated scenario in which integration time does *not* increase as luminance decreases (i.e. integration time observed at the highest background tested is held constant).

DOI: https://doi.org/10.7554/eLife.38281.022

The following source data is available for figure 8:

**Source data 1.** Excel spreadsheet with data for *Figure 8A*.

DOI: https://doi.org/10.7554/eLife.38281.023

to the situation with a fixed short integration time. The relative magnitude of the statistical fluctuations in the input decreases as light levels increase, and the commensurate decrease in integration time enables sensitivity to higher frequency inputs (e.g. *Figure 5—figure supplement 1*).

The speeding of rod signals has been attributed to a shift in routing of rod signals from the (presumed slow) primary pathway to the (presumed fast) secondary or cone bipolar pathway. Experiments in non-primate retinas, particularly mice, provide at least partial support for this hypothesis (*Soucy et al., 1998*; *Deans et al., 2002*; *Trexler et al., 2005*). The work described here shows, unexpectedly, that rod-derived signals in primate retinal ganglion cells are restricted to the primary pathway across light levels and across a broad range of stimuli. This implies that light-level-dependent changes in rod signaling in primate, such as those illustrated in *Figure 8*, are not due to a change in routing, but instead to flexibility within the primary pathway. Indeed, we find that the speeding of rod signals in the retinal output can be explained by a change in kinetics of signals in the rods themselves – with approximately equal contributions from phototransduction and active conductances in the inner segment.

## Segregation of rod and cone signals

Recent anatomical and functional studies highlight differences in the segregation of rod and cone signals between rodent and primate retina. Specifically, mouse rod bipolar cells, long thought to receive input exclusively from rods, make some contacts with cones (*Behrens et al., 2016*) and can convey cone-derived signals (*Pang et al., 2010*; *Szikra et al., 2014*). Cone inputs to rod bipolar cells

have not been observed in primate retina. Similarly, rod contacts on cone bipolar dendrites appear more numerous in mouse than primate (*Tsukamoto and Omi, 2014*; *Tsukamoto and Omi, 2016*). This suggests a greater mixing of rod and cone signals in rodent retina than in primate retina.

Our work here shows a surprisingly clear separation of rod and cone signals in primate retina – with even the well-established gap junctions between rods and cones (*Kolb, 1977*; *Hornstein et al., 2005*) conveying at most a small rod-derived signal. The resulting separation of rod and cone signals through most of the retinal circuitry maximizes opportunities for independent processing. Such independent processing may be particularly important under mesopic conditions when rod signals are approaching saturation and cone signals are small and threatened by noise. Our work also provides a cautionary reminder that mechanistic studies derived from mouse may not be directly applicable to humans.

## Potential limitations

All the experiments described here used an isolated, in vitro retina preparation. Several factors could cause responses in this preparation to differ from in vivo conditions. An issue of particular concern is the washout of neuromodulators that could enhance rod signaling through the secondary pathway – for example the dopaminergic signaling important in circadian rhythms (*Ribelayga et al., 2008*). We note, however, that the same preparation does not eliminate the secondary pathway in mouse, but this issue is confounded by a lack of circadian regulation in C57/Bl6 mice (*Jin and Ribelayga, 2016*). Another issue is damage to the retina. We attempted to minimize this issue by adopting strict criteria for including cells in our data set (see Materials and methods), based on the assumption that the most sensitive preparations we see most closely resemble in vivo conditions.

Rod suction and RGC recordings (*Figures 2* and *4*) indicate that rods are largely saturated at light levels at or above ~300 R*/rod/s in primates and ~1200 R*/rod/s in mice. Our physiological estimate of the upper end of primate rod signaling agrees well with human perceptual results (*Sharpe et al., 1992*). However, recordings from rodents reveal that rods can modulate retinal outputs at considerably higher light levels (*Yin et al., 2006*; *Naarendorp et al., 2010*). These remaining rod responses rely at least in part on a form of bleaching adaptation that allows rods to regain sensitivity after prolonged exposure (>10 min) to photopic backgrounds (*Tikidji-Hamburyan et al., 2017*). On short time scales (≤10 min) and consistent with our results (*Figure 4*), Tikidji-Hamburyan and colleagues found that rods could no longer modulate RGC output at backgrounds at or above $10^4$ R*/rod/s. It will be interesting to see if this form of slow adaptation found in mouse rods is also present in primates. We did not probe adaptation on time scales longer than 2–3 min as we wanted to minimize bleaching given the absence of photopigment regeneration in our preparations.

## Rod flicker null

We did not find evidence for the 15 Hz flicker null that has been observed perceptually (*Sharpe et al., 1989*; *Sharpe and Stockman, 1999*). This flicker null was similarly not observed in previous in vivo recordings from monkey parasol and midget RGCs (*Cao et al., 2010*). Several issues could contribute to this difference: (1) Cancellation of rod signals was proposed to occur between the primary and secondary pathways, but this might be incorrect. Alternatively, cancellation could reflect the downstream integration of 15 Hz signals from On and Off pathways, or it might arise in a distinct set of RGCs from those recorded here. (2) The rod flicker null is observed over a narrow luminance range that may have been missed in our experiments. (3) Neuromodulators that enhance rod-cone coupling are washed out in our in vitro preparation. Future experiments will be needed to uncover the origins of this perceptual effect.

## Linking neural circuits to perception

Interactions between rod and cone signals affect many aspects of mesopic vision (reviewed by (*Buck, 2014*; *Stockman and Sharpe, 2006*). The experiments described here indicate that the speeding of rod-derived signals is not due to a change in routing, as often assumed, but instead occurs largely within the rods themselves. Thus, this work implicates rod adaptation, and not rerouting, as the critical retinal mechanism that shapes the time course of rod-derived signals and ultimately human perception under scotopic and mesopic conditions.

## Materials and methods

### Electrophysiology

Experiments were conducted on whole mount or slice (200 μm thick) preparations of primate ( M. fascicularis, M. mulatta and M. nemestrina) or mouse (C57/BL6) retina as previously described (*Dunn et al., 2007*; *Trong and Rieke, 2008*). Retinas were obtained through the Tissue Distribution Program of the Washington National Primate Research Center; all procedures followed protocols approved by the Institutional Animal Care and Use Committee at the University of Washington. In brief, pieces of retina attached to the pigment epithelium were stored in ~32–34° C oxygenated (95% $O_2$/5% $CO_2$) Ames medium (Sigma) and dark-adapted for >1 hr. Pieces of retina ($\geq$15° eccentricity for primate) were then isolated from the pigment epithelium under infrared illumination and either flattened onto poly-L-lysine slides (whole mount: cone, horizontal cell, AII amacrine cell and RGC recordings) or embedded in agarose and sliced (rod current clamp recordings). Once under the microscope, tissue was perfused with Ames medium at a rate of ~8 mL/min. Most data was collected during the nominal circadian day for both species.

Extracellular spike recordings from On parasol retinal ganglion cells used ~3 MΩ electrodes containing Ames medium. Voltage-clamp whole-cell recordings used electrodes (RGC: 2–3 MΩ, AII, HC: 5–6 MΩ) containing (in mM): 105 Cs methanesulfonate, 10 TEA-Cl, 20 HEPES, 10 EGTA, 2 QX-314, 5 Mg-ATP, 0.5 Tris-GTP and 0.1 Alexa (488) hydrazide (~280 mOsm; pH ~7.3 with CsOH). To isolate excitatory synaptic input, cells were held at the estimated reversal potential for inhibitory input (~−60 mV). This voltage was adjusted for each cell to maximize isolation. Current-clamp whole-cell recordings were conducted with electrodes (AII: 5–6 MΩ, HC: 5–6 MΩ, rods, cones: 10–12 MΩ) containing (in mM): 123 K-aspartate, 10 KCl, 10 HEPES, 1 $MgCl_2$, 1 $CaCl_2$, 2 EGTA (omitted for rod and cone recordings), 4 Mg-ATP, 0.5 Tris-GTP and 0.1 Alexa (488) hydrazide (~280 mOsm; pH ~7.2 with KOH). Absolute voltage values have not been corrected for liquid junction potentials ($K^+$-based = −10.8 mV; $Cs^+$-based = −8.5 mV). In initial whole-cell experiments, cell types were confirmed by fluorescence imaging following recording.

Perforated patch clamp recordings from cone photoreceptors were performed in current clamp using electrodes (9–11 MΩ) containing (in mM): 115 potassium aspartate, 1 $MgCl_2$, 4 KCl, 10 HEPES, 10 diTris phosphocreatine hydrate, 4 Mg-ATP, 0.5 Tris-GTP (276–278 mOsm, pH 7.1–7.15 with KOH). Gramicidin was added to the internal solution at 30 μg/mL. Upon sealing on a cell, access was monitored by tracking the membrane potential as well as the response amplitude to a constant amplitude probe flash. Once access equilibrated (5–25 min), recordings were started. Throughout perforated patch recordings, the membrane potential and response to a reference flash were monitored to ensure that electrical access to the cell remained stable.

For suction recordings, a suspension of finely chopped retina was transferred to a recording chamber, pieces of retina were briefly allowed to settle, then superfused (2–3 ml/min, 32 ± 1°C). Suction electrodes (3–4 MΩ, tip inner diameter of ~1.6 μm) were filled with HEPES-buffered Ames and were voltage clamped at 0 mV (*Field and Rieke, 2002a*). Individual rod outer segments were drawn into the recording pipette under gentle suction and selected for extended recordings if their outer segment had not been obviously damaged by the suction procedure and if they had maximal light responses exceeding 20 pA for primate and 8 pA for mouse.

Activity of mGluR6 receptors expressed by On bipolar cells was suppressed in some experiments using a mixture of LY341495 (2.5 μM) and APB (7.5 μM). This approach was chosen to suppress modulated responses of On bipolar cells while minimizing perturbations to the state of the retina associated with tonic changes in mGluR6 receptor activity (*Ala-Laurila et al., 2011*; *Grimes et al., 2014a*). In mouse rod bipolar cells and AII amacrine cells, this mixture of drugs maintained the resting potential and suppressed light-dependent responses by >90% (data not shown). Incomplete block of mGluR6 activity would lead to an overestimate of contributions from non-primary pathways.

### Cell selection criteria

To assay the overall health and sensitivity of the retina, we measured On RGC responses (On parasol cells in primate and On Alpha RGCs in mouse) to brief test flashes. Recorded cells met several criteria: backgrounds producing ~0.01 R*/rod/s elicited a clear increase in firing rate (from near 0 in darkness to 2–5 spikes/s); flashes producing 0.02 R*/rod at a background of 0.3 R*/rod/s elicited clear

responses; and, in primate, rod- and cone-derived responses were clearly separable using 405 and 640 nm lights at a background of 10–20 R*/rod/s. We measured RGC sensitivity in every piece of retina, and only proceeded to collect data when these criteria were met. Approximately half of the preparations met these criteria.

## Visual stimuli

Psychophysical measurements of rod vision in humans indicate that dark adaptation is near complete in 30–45 min (*Sharpe et al., 1989*). Thus, isolated primate retinas (attached to the pigment epithelium) were dark-adapted for ≥1 hr before any data was collected. Mice were dark adapted for ≥12 hr before isolating the retina. As we progressed from darkness to mesopic conditions, new backgrounds were typically applied for ≥1 min before data was collected. This approach was consistent for all preparations and cell-specific recordings. Responses to repeated stimuli delivered ~1 min and ~3 min after a change in background did not differ noticeably.

Stimuli were presented and data acquired using a custom written stimulation and acquisition software package (*Symphony-DAS, 2018*: http://symphony-das.github.io/). Full field illumination (diameter: 500–560 μm) was delivered to the preparation through a customized condenser from three LEDs (peak power at 405, 520 and 640 nm). Rod responses were elicited by 405 nm light and cone responses by 640 nm light for all but the control experiments described in the next paragraph. Light intensities (photons/μm²/s) were converted to photoisomerization rates (R*/photoreceptor/s) using the estimated collecting area of rods and cones (1 and 0.37 μm², respectively, for primate; 0.5 μm² for mouse rods), the LED emission spectra and the photoreceptor absorption spectra (*Baylor et al., 1984*; *Baylor et al., 1987*). Rod- and cone-preferring flashes were 10 ms in duration.

The dim 405 nm light stimulated RGCs primarily through rods rather than cones. To test for a cone contribution to these responses, we monitored the ratio of On parasol responses elicited by the 520 and 405 nm LEDs. This ratio is predicted to differ by a factor of ~2.4 for responses elicited by rods and M cones based on the measured LED spectral output and the rod and M cone spectral sensitivities (On parasol cells do not get significant S cone input [*Field et al., 2010*]). The ratio increased noticeably at 100–300 R*/rod/s, reaching a factor of 1.8 at 300 R*/rod/s. Together with the theoretical factor of 2.4, this indicates that responses to the 520 nm LED represented a ~60% contribution from cones and ~40% from rods at 300 R*/rod/s. Together with the higher relative sensitivity of the rods compared to M cones to 405 nm light, this indicates that ~25% of the response to the 405 nm LED at 300R*/rod/s originated in cones. Cone contributions were substantially smaller at lower background light levels. Note that we used the 520 nm LED only for this control experiment, and otherwise used the 405 and 640 nm LEDs.

## Analysis

For spike recordings from ganglion cells, we detected spike times and compiled them into peristimulus time histograms as previously described (*Murphy and Rieke, 2006*). Assuming linearity, rod responses to sinusoidal stimuli (*Figure 7*) were predicted by convolving the measured flash responses with sinusoids of the various frequencies. Electrophysiology example traces presented throughout the figures represent the average of 5–20 raw responses to the same stimuli. Data are presented as mean ± SEM using an unbiased estimate of the standard deviation. We did not have sufficient examples to check whether the data was normally distributed.

## Acknowledgements

We thank Cole Graydon for commenting on the manuscript, Chris English for assistance with tissue acquisition and Shellee Cunningham for general technical assistance. Support was provided by HHMI and NIH (EY028111).

## Additional information

### Competing interests

Fred Rieke: Reviewing editor, *eLife*. The other authors declare that no competing interests exist.

## Funding

| Funder | Grant reference number | Author |
| --- | --- | --- |
| National Institutes of Health | EY028111 | Fred Rieke |
| Howard Hughes Medical Institute | | Fred Rieke |

The funders had no role in study design, data collection and interpretation, or the decision to submit the work for publication.

## Author contributions

William N Grimes, Fred Rieke, Conceptualization, Data curation, Formal analysis, Investigation, Writing—original draft, Writing—review and editing; Jacob Baudin, Anthony W Azevedo, Data curation, Investigation, Writing—review and editing

## Author ORCIDs

Fred Rieke http://orcid.org/0000-0002-1052-2609

## Ethics

Animal experimentation: We obtained primate retinas (Macaca fascicularis, Macaca nemestrina and Macaca mulatta of either sex, ages 3-19 years) through the Tissue Distribution Program of the Regional Primate Research Center. All protocols were approved by the Institutional Animal Care and Use Committee at the University of Washington (protocol 4140-01).

## Decision letter and Author response

Decision letter https://doi.org/10.7554/eLife.38281.026
Author response https://doi.org/10.7554/eLife.38281.027

# Additional files

## Supplementary files

• Transparent reporting form
DOI: https://doi.org/10.7554/eLife.38281.024

We have provided source data for the population analysis for all the main figures (as Excel files) and the raw traces from Figure 2 (Figure 2—source data 2 and Figure 2—source data 3).

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
