## [Decision Letter]

Thank you for submitting your article "Rod signaling in primate retina: range, routing and kinetics" for consideration by *eLife*. Your article has been reviewed by three peer reviewers, and the evaluation has been overseen by a Reviewing Editor and Timothy Behrens as the Senior Editor. The following individual involved in review of your submission has agreed to reveal his identity: Markus Meister (Reviewer #2).

The reviewers have discussed the reviews with one another and the Reviewing Editor has drafted this decision to help you prepare a revised submission.

Summary:

This manuscript presents new insights on visual processing in the retina, specifically how signals from rod photoreceptors are handled. Our visual system must operate over a huge range of light intensities, about 9 log units in the course of a day. In adaptation to this challenge the retina uses two kinds of photoreceptors: In the dimmest conditions only the sensitive rods are active, in the brightest conditions only the cones. In between the retina gradually switches from one input neuron to the other. However, even before the cones take over, the rod pathway undergoes substantial changes with increasing light level: the gain decreases and the speed of processing increases. This article challenges the prevailing notion of how those changes are accomplished.

The authors distinguish three routes: (1) an interneuron pathway through specialized rod bipolar and amacrine cells, which allows for high spatial convergence and high gain; (2) direct transmission of rod signals to cones via electrical junctions; and (3) transmission from rods to Off bipolar cells that predominantly handle cone signals. Each of these pathways imposes different gain and kinetics on the rod signal because different synaptic pathways are engaged.

With this background it has been proposed that the changes in human perception of rod stimuli that occur with increasing light level reflect changes in the neural routing of signals through the different synaptic pathways. In the mouse retina there is reasonable evidence for this, but the idea has received less scrutiny in primate retina, which ultimately matters for human vision. Here the authors come to the surprising conclusion that the primate retina uses pathway 1 for rod signals almost exclusively. Furthermore they claim that the changes in kinetics of rod signal processing observed in human psychophysics are largely explained by changes occurring within the rod photoreceptor cell itself. Parallel experiments in mouse retina using the same strategy gave very different results: a large contribution from the pathways 2 and 3 over the functional rod range.

Essential revisions:

1) The authors could improve the comparison to existing data in mouse. For example, Ke et al., 2014 reported that responses in ganglion cells and AII amacrine cells showed a strong reduction in the presence of DNQX at ~250-300 R*/rod/s. Those results would suggest that, similar to the primate recordings in the present study, the primary rod pathway makes the major contribution also in mouse at a relatively high background level. That makes the mouse circuits seem more similar to the primate circuits than the comparison presented in the current study. Was there a difference between the results presented here and in the earlier work?

The Ke et al. study also showed that ganglion cell responses in control conditions were not slower than those in DNQX (Figure 5I/J of that paper) – suggesting that moving from the primary+secondary to just the secondary pathway did not affect kinetics much, consistent with the present results in primate apparently. That again makes the mouse and primate results seem more similar than presented in the current study.

The key point here is that it would be good either in Introduction or Discussion to make a more substantial comparison to the findings in Ke et al., 2014 – where the main point was that the mouse primary rod pathway was dominating at light levels higher than those believed previously to be carried by this pathway.

Furthermore, Grimes et al., 2014 suggested that the primary rod pathway can become suppressed and shut off in bright light because the rod bipolar cell synapses becomes suppressed, causing hyperpolarization of the AII, etc. Does that seem to happen in the primate circuit? What happens to the AII membrane potential in background light (Figure 3)?

2) The paper does not present a clear role for synaptic transmission in shaping the gain or kinetics of signals in the primate retina. For example, Oesch and Diamond, 2011 describe a synaptic mechanism for adaptation in rod bipolar cells. How do those results relate to e.g., the Weber adaptation described in the primary pathway in the current study?

It is also not clear how synaptic transmission could play a role in the kinetics of rod responses. For example, rods apparently hyperpolarize 9 mV between 1 and 60 R*/rod/s (Figure 7). This hyperpolarization affects the kinetics of the voltage response. What about the presumed affect, if any, at the synapse?

3) The comparison of mouse and primate rod recordings (Figure 4A) depends on the estimate of saturation in these curves. But there are not many points showing an unambiguous asymptote in either case. There is some concern that the comparison depends too much on the normalization of each curve.

Figure 4B leaves the reader with the impression that mouse AII recordings in NBQX are saturated at 100 R*/rod/s, but there is a clear response to contrast at 250-300 R*/rod/s in AIIs and postsynaptic ganglion cells under similar conditions reported previously (Ke et al., 2014). There should be a little more context when comparing the two species and providing an impression of when certain cell types or pathways are 'shut down' and can be safely assumed to no longer contribute to circuit function at a given background.

4) Psychophysics of parallel rod pathways. The authors suggest that the human perceptual effects can be explained by a single pathway for rod signals whose gain and kinetics vary as a function of light level. However, some psychophysical experiments show that multiple pathways for rod signals are in operation simultaneously, see for example the phenomenon of "rod self-cancellation" (Stockman and Sharpe, 2006). Can that be reconciled with the single-pathway hypothesis?

5) Health of rod-cone junctions. In terms of neural mechanisms, the authors suggest that in primate retina the rod-cone junctions are too weak to initiate a substantial pathway 2. Is it possible that these junctions are altered by the procedure of isolating the retina in vitro? The junctions seem to be sensitive to modulation, for example by circadian time, through the actions of dopamine (Ribelayga et al., 2008). That could be perturbed by isolating the retina, which would directly affect the main conclusion. How could one check whether these junctions are in the same state as in the intact human eye?

6) Strength of the third pathway. The single pathway conclusion seems less compelling for the Off retinal ganglion cells. Suppressing the On bipolars (which blocks pathway 1) eliminates 80% of the rod signals at a light level of 20R*/s, but only 20% at 200 R*/s (subsection “Routing of rod signals in Off retinal circuits”, fourth paragraph). In between there is a large range and it all falls within the 300R*/s range that the authors consider for rod signaling. Thus it seems that there could be a substantial contribution from pathways 2 or 3 over a substantial part of the rod signaling range. This should be elaborated further.

7) A fourth pathway. Another potential pathway for rod signals was not considered here: rods excite H1 horizontal cells which inhibit cone terminals. There is reasonable evidence for this route in the mouse retina, where it seems to provide the entire receptive field surround of certain ganglion cells (Joesch and Meister, 2016; Szikra et al., 2014; Trümpler et al., 2008). This may affect the interpretation of some of the mouse experiments here. In the primate retina the H1 cell seems to be much less sensitive to direct rod input (Verweij et al., 1999), as stated in the manuscript (subsection “Rod signal routing”, fifth paragraph).

8) Presumably, the macaque and mouse retinas are being examined at opposite phases of their circadian rhythm. Does this matter for the comparison of their rod pathways?

9) The cell selection criteria are strong for the primate retina. It would be helpful to mention if there were any selection criteria for the mouse retina, or if it can be safely assumed that all mouse retinas are in excellent condition given how they are collected.

10) Were the background lights applied for the same duration across all experiments? How much time elapsed from the application of the background to the delivery of flashes? How do the durations of the backgrounds used here compare to those used in psychophysical experiments probing rod vision? These points should be made clear.

11) Figure 8. Why compare the time to peak of rods (8A) with the integration time of RGCs (8B) rather than the integration time for both? For 8C, is the x-axis the RGC integration time? How was the integration time measured (does it include the undershoot in the RGC response)? How was SNR calculated? How was the simulation done? Explanation of the contents of this figure needs to be greatly expanded. As presented, I have a hard time evaluating them.

*Additional comment from Reviewer 3:*

One additional comment: It would be helpful if the authors clarified the wavelength of their cone-preferring stimuli. It's only mentioned for Figure 3—figure supplement 2 (640 nm) but the Materials and methods makes it sound like they focused on 520 nm. As reviewer #2 mentioned (comment 7), 520 does not distinguish between rods and M cones well.

I used Govardovskii nomograms (A1, β bands included) to put some numbers on this. The ratio of absorption for monochromatic 520- vs. 405-nm light is 4 for rods, 5.2 for M cones, and 3.9 for L cones-not very different. Between rods and cones, the ratio of ratios is 1.3. The authors give 2.4 in their Materials and methods (subsection “Visual stimuli”). Part of the discrepancy may be due to their use of more broadband light, but this point should be clarified.

For cone-preferring stimuli, 640 is better (640/405 being 0.004 for rods, 0.1 for M cones, and 0.7 for L cones).

---

## [Author Response]

Essential revisions:1) The authors could improve the comparison to existing data in mouse. For example, Ke et al., 2014 reported that responses in ganglion cells and AII amacrine cells showed a strong reduction in the presence of DNQX at ~250-300 R*/rod/s. Those results would suggest that, similar to the primate recordings in the present study, the primary rod pathway makes the major contribution also in mouse at a relatively high background level. That makes the mouse circuits seem more similar to the primate circuits than the comparison presented in the current study. Was there a difference between the results presented here and in the earlier work?The Ke et al. study also showed that ganglion cell responses in control conditions were not slower than those in DNQX (Figure 5I/J of that paper) – suggesting that moving from the primary+secondary to just the secondary pathway did not affect kinetics much, consistent with the present results in primate apparently. That again makes the mouse and primate results seem more similar than presented in the current study.The key point here is that it would be good either in Introduction or Discussion to make a more substantial comparison to the findings in Ke et al., 2014 – where the main point was that the mouse primary rod pathway was dominating at light levels higher than those believed previously to be carried by this pathway.

These are excellent points, and our original treatment of the mouse work was too superficial. We have revised the paper in several places, including both the Introduction and Discussion, to clarify that both Ke et al., 2014 and Grimes et al., 2014 find only a partial rerouting of rod signals – not the complete switch from one pathway to another suggested by the routing hypothesis. We further clarify in the Introduction that there is no physiological evidence for fast and slow rod pathways. We did not specifically mention the similarity in kinetics observed in Ke et al., 2014 because of the reliance of those results on DNQX and the possibility that it could have unanticipated effects on response kinetics (e.g. due to block of inhibitory circuits). These issues are covered most completely in the (new) next to last paragraph of the Introduction.

Several studies in mouse have produced evidence for the involvement of multiple rod pathways, although details about the relative strengths of the different pathways differ across studies. The involvement of multiple pathways itself, however, differs from what we see in primate. To avoid a comparison of results using somewhat different recording conditions and stimuli, we repeated key experiments using both mouse and primate retina. Our results show that the secondary pathway makes a much larger contribution to signaling in mouse compared to primate retina for identical stimulus conditions. To better emphasize our focus here, we have changed the wording of the initial sentence in the section on primate vs. mouse comparisons: ‘The lack of secondary pathway contributions in primate was unexpected, as previous experiments in rodents have assigned a substantial role to the secondary pathway in transmitting rod signals.’

Furthermore, Grimes et al., 2014 suggested that the primary rod pathway can become suppressed and shut off in bright light because the rod bipolar cell synapses becomes suppressed, causing hyperpolarization of the AII, etc. Does that seem to happen in the primate circuit? What happens to the AII membrane potential in background light (Figure 3)?

The effect seen in Grimes et al., 2014 was a partial shut off of the rod bipolar pathway (at most ~50% of the signal in the AII was insensitive to NBQX in that study – suggesting an approximately equal distribution of signals across primary and secondary pathways at high rod light levels). Our AII recordings from primate do not show a reliable trend in membrane potential as a function of luminance like that observed in mouse. We similarly have not seen a clear change in nonlinear spatial integration in On parasol cells like that reported in Grimes et al. in mouse On Α cells. We now note the apparent absence of AII hyperpolarization in primate in the text (subsection “Rod signal routing”, third paragraph).

2) The paper does not present a clear role for synaptic transmission in shaping the gain or kinetics of signals in the primate retina. For example, Oesch and Diamond, 2011 describe a synaptic mechanism for adaptation in rod bipolar cells. How do those results relate to e.g., the Weber adaptation described in the primary pathway in the current study?

Thank you – this is a point about which we should have been clearer. We have expanded the comparison of rod and ganglion cell gain in Figure 2 and the associated text. First, we added a Weber line (slope -1 on a log-log plot, indicating an inverse scaling of gain with background) to the gain data in Figure 2D, showing that both rod and ganglion cell gain follow Weber behavior across a range of light levels. We have clarified that the reduction in ganglion cell gain at lower light levels likely originates at the rod bipolar output synapse (subsection “Rod signaling range and saturation”, third paragraph) and cited the Oesch and Diamond paper there. We hesitate to get into this issue too much further since our focus here is on which circuits convey rod signals and not adaptation. Several previous studies have compared adaptation at different stages of the rod bipolar pathway and used these to constrain the relative roles of different circuit components in controlling RGC gain. We cite these now in the text in discussing rod vs. post-rod gain controls.

It is also not clear how synaptic transmission could play a role in the kinetics of rod responses. For example, rods apparently hyperpolarize 9 mV between 1 and 60 R*/rod/s (Figure 7). This hyperpolarization affects the kinetics of the voltage response. What about the presumed affect, if any, at the synapse?

This is also interesting. Indeed, we were quite surprised to find how well the change in rod kinetics accounted for the change in ganglion cell kinetics (Figure 7F). This does not mean that synaptic transmission or other post-rod mechanisms do not contribute to shaping the kinetics of ganglion cell responses, but the bulk of that timing shift appears to occur in the rods themselves. We now note this at the end of the text describing Figure 7 (subsection “Rod signal kinetics”, last paragraph).

3) The comparison of mouse and primate rod recordings (Figure 4A) depends on the estimate of saturation in these curves. But there are not many points showing an unambiguous asymptote in either case. There is some concern that the comparison depends too much on the normalization of each curve.

These curves are normalized by the current suppressed by a brief saturating flash – which gives a much more reliable measure of the full dark current than a step. This is indicated in Figure 2A, and we have clarified in the text (subsection “Comparison of rod signal routing in mouse and primate”, second paragraph) that the same normalization was applied to the mouse data. This was not as clear as it should have been, especially in the text discussing Figure 4. This normalization means that the scaling of the step responses in Figure 2A and Figure 4A does not depend on estimating saturation from the data plotted, but instead the peak current suppression is well constrained independently of the step response data. The key point of this comparison for the present paper is that there is a large range of light levels in mouse over which substantial signals are conveyed through the secondary and tertiary pathways and rods are not saturated. We have revised Figures 4 and 6 help make this point more clearly.

Figure 4B leaves the reader with the impression that mouse AII recordings in NBQX are saturated at 100 R*/rod/s, but there is a clear response to contrast at 250-300 R*/rod/s in AIIs and postsynaptic ganglion cells under similar conditions reported previously (Ke et al., 2014). There should be a little more context when comparing the two species and providing an impression of when certain cell types or pathways are 'shut down' and can be safely assumed to no longer contribute to circuit function at a given background.

The responses in Figure 4B are to fixed contrast flashes across a range of backgrounds. Thus a value of 1 does not equate to saturation of a pathway, but instead argues that the pathway obeys Weber adaptation over this range (i.e. the signal gain is inversely proportional to the background). As such, this data does not provide insight into the state of the primary pathway (e.g. shut down, not shut down); instead, it shows how signals through the secondary pathways emerge at these light levels. We have revised the text to clarify this issue, stating “For backgrounds ≥ 50 R*/rod/s, responses of mouse horizontal cells and NBQX-insensitive responses of mouse AII amacrines to fixed-contrast stimuli depended only weakly on background. This does not mean that responses are saturated; instead, consistent responses to fixed contrast across backgrounds is indicative of Weber behavior (i.e. that gain scales inversely with luminance).” We have also updated the Figure 4 legend to clarify this issue. This issue also comes up in Figure 3, in which responses are similarly scaled; we have revised text describing that figure with the same issues in mind (see subsection “Rod signal routing”, fourth paragraph).

4) Psychophysics of parallel rod pathways. The authors suggest that the human perceptual effects can be explained by a single pathway for rod signals whose gain and kinetics vary as a function of light level. However, some psychophysical experiments show that multiple pathways for rod signals are in operation simultaneously, see for example the phenomenon of "rod self-cancellation" (Stockman and Sharpe, 2006). Can that be reconciled with the single-pathway hypothesis?

We have looked for but failed to find any evidence for rod self-cancellation in our recordings from primate retina (despite seeing beautiful rod-cone cancellation at 7.5 Hz). We now include these measurements in Figure 5—figure supplement 1. We have added a section to the Results describing these experiments (subsection “Routing of responses to continuous stimuli”, last paragraph), as well as a new Discussion section (subsection “Rod flicker null”).

Stockman and Sharpe, 1999 mention alternative mechanistic hypotheses that could also explain these results, and we summarize those in the Discussion (new “Rod flicker null” subsection). We also note that we may wash out important neuromodulators that enhance rod-cone coupling, although this does not appear to be the case in identical preparations of mouse retina in which rod-cone coupling and secondary pathway signaling is clear (subsection “Potential limitations”, first paragraph).

5) Health of rod-cone junctions. In terms of neural mechanisms, the authors suggest that in primate retina the rod-cone junctions are too weak to initiate a substantial pathway 2. Is it possible that these junctions are altered by the procedure of isolating the retina in vitro? The junctions seem to be sensitive to modulation, for example by circadian time, through the actions of dopamine (Ribelayga et al., 2008). That could be perturbed by isolating the retina, which would directly affect the main conclusion. How could one check whether these junctions are in the same state as in the intact human eye?

The health of in vitro retinal preparations is always a concern, and we cannot rule out the possibility that we have washed out important neuromodulators or otherwise altered function when isolating the retina. We have clarified our selection criteria for retinal sensitivity, noted in the Materials and methods subsection “Cell selection criteria”. This rests on the assumption, now stated, that the most sensitive responses we observe closely reflect in vivo conditions. These criteria lead to consistent experimental results in our hands. The rod-cone gap junctions are functional in our experimental conditions because we can reliably record rod signals (albeit weak ones) in both horizontal cells and cones. Furthermore, for our mouse experiments, in which the secondary pathway is clearly intact, we isolated and stored the retina following the same procedures as in our primate experiments, and those procedures do not eliminate the secondary pathway in mouse. We have now added a section to the Discussion where we mention circadian rhythms and the possibility of neuromodulator washout and retina damage (subsection “Potential limitations”, first paragraph).

6) Strength of the third pathway. The single pathway conclusion seems less compelling for the Off retinal ganglion cells. Suppressing the On bipolars (which blocks pathway 1) eliminates 80% of the rod signals at a light level of 20R*/s, but only 20% at 200 R*/s (subsection “Routing of rod signals in Off retinal circuits”, fourth paragraph). In between there is a large range and it all falls within the 300R*/s range that the authors consider for rod signaling. Thus it seems that there could be a substantial contribution from pathways 2 or 3 over a substantial part of the rod signaling range. This should be elaborated further.

An important issue in interpreting these experiments, which we neglected to discuss, is that the relative strength of rod- and cone-derived responses in On and Off ganglion cells differs. In particular, rod-derived responses are about 2-fold weaker relative to cone-derived responses in Off cells than On cells. This means that cone-derived responses should become relatively prominent at lower light levels in Off cells – which is exactly what we observe. We now note these issues in the text (subsection “Routing of rod signals in Off retinal circuits”, fifth paragraph). We have also revised the text earlier (subsection “Rod signaling range and saturation”, last paragraph) to clarify that 300 R*/rod/s is simply the light level at which, for On parasol cells, the balance of signaling appears to shift in favor of cones over rods, but that shift depends on the relative size of rod and cone signals and hence could occur at different light levels for different circuits.

7) A fourth pathway. Another potential pathway for rod signals was not considered here: rods excite H1 horizontal cells which inhibit cone terminals. There is reasonable evidence for this route in the mouse retina, where it seems to provide the entire receptive field surround of certain ganglion cells (Joesch and Meister, 2016; Szikra et al., 2014; Trümpler et al., 2008). This may affect the interpretation of some of the mouse experiments here. In the primate retina the H1 cell seems to be much less sensitive to direct rod input (Verweij et al., 1999), as stated in the manuscript (subsection “Rod signal routing”, fifth paragraph).

We now explicitly mention this possibility in the Introduction (second paragraph) and in Results where we first describe horizontal cell recordings (subsection “Rod signal routing”, fifth paragraph). We do not feel this possibility requires reinterpretation of our results, as signaling from rods to horizontal cells to cones or cone bipolar cells would contribute to the strength of rod signals reaching RGCs via routes other than the primary pathway. Since the primary pathway seems to dominate in primate, none of the alternative pathways appear to make large contributions to RGC responses.

8) Presumably, the macaque and mouse retinas are being examined at opposite phases of their circadian rhythm. Does this matter for the comparison of their rod pathways?

The light-dark cycle was similar for mice and monkeys used in these studies (in both cases experiments began during the nominal day). Our primate experiments often continued for >12 hours without noticeable changes in the balance of rod and cone signals as measured in ganglion cells. Recent work has shown that gap junctional coupling in C57/Bl6 mice is not regulated by circadian rhythms. We now mention the lack of circadian regulation in C57/Bl6 and the time of day of our recordings (subsection “Potential limitations”, first paragraph and subsection “Electrophysiology”, first paragraph).

9) The cell selection criteria are strong for the primate retina. It would be helpful to mention if there were any selection criteria for the mouse retina, or if it can be safely assumed that all mouse retinas are in excellent condition given how they are collected.

We have added a section to the Materials and methods (subsection “Cell selection criteria”) detailing cell selection in both cases. It is essential for both mouse and primate and is an issue we try to be quite systematic about. There is certainly substantial variability in the sensitivity of mouse retina, and thus selection criteria are needed for mouse too.

10) Were the background lights applied for the same duration across all experiments? How much time elapsed from the application of the background to the delivery of flashes? How do the durations of the backgrounds used here compare to those used in psychophysical experiments probing rod vision? These points should be made clear.

Thanks for raising this issue. We have now clarified both the duration that we dark adapt the retina prior to experiments and the duration of background exposures prior to data collection. Both are compared to behavioral experiments. We did not begin data collection until at least 1 min after the onset of a background and checked for stability of responses at ~1 and ~3 min. This timing was used for all experiments in the paper. This is now described in the Results (subsection “Rod signaling range and saturation”, second and third paragraphs), Materials and methods (subsection “Visual Stimuli”, first paragraph), and at the end of the (new) “Potential Limitations” subsection in the Discussion.

11) Figure 8. Why compare the time to peak of rods (8A) with the integration time of RGCs (8B) rather than the integration time for both? For 8C, is the x-axis the RGC integration time? How was the integration time measured (does it include the undershoot in the RGC response)? How was SNR calculated? How was the simulation done? Explanation of the contents of this figure needs to be greatly expanded. As presented, I have a hard time evaluating them.

We apologize for any confusion. The analysis in Figure 8A and B was based entirely on RGC signals (not rods). To simplify this analysis and avoid confusion we have removed the measures of time to peak (previous Figure 8A) and expanded the figure legend to define the integration time and the model design. We have also expanded the discussion of this figure in the main text considerably (Discussion, new third paragraph).

Addition comment from Reviewer 3:One additional comment: It would be helpful if the authors clarified the wavelength of their cone-preferring stimuli. It's only mentioned for Figure 3—figure supplement 2 (640 nm) but the Materials and methods makes it sound like they focused on 520 nm. As reviewer #2 mentioned (comment 7), 520 does not distinguish between rods and M cones well.I used Govardovskii nomograms (A1, β bands included) to put some numbers on this. The ratio of absorption for monochromatic 520- vs. 405-nm light is 4 for rods, 5.2 for M cones, and 3.9 for L cones-not very different. Between rods and cones, the ratio of ratios is 1.3. The authors give 2.4 in their Materials and methods (subsection “Visual stimuli”). Part of the discrepancy may be due to their use of more broadband light, but this point should be clarified.For cone-preferring stimuli, 640 is better (640/405 being 0.004 for rods, 0.1 for M cones, and 0.7 for L cones).

We clarified that we used 640 nm light to selectively activate cones (see subsection “Visual stimuli”, second paragraph). The one exception is the experiment to check for cone responses to 405 nm light. The spectral output of the 405 nm LED is fairly skewed towards long wavelengths, and that skew accounts for the difference in rod/cone ratios predicted for monochromatic vs broad-band stimuli. We now note that the estimated ratio incorporates the measured LED spectrum.